# Effect of Processing Time of Steam-Explosion for the Extraction of Cellulose Fibers from *Phoenix canariensis* Palm Leaves as Potential Renewable Feedstock for Materials

**DOI:** 10.3390/polym14235206

**Published:** 2022-11-29

**Authors:** Maria Angeles Pérez-Limiñana, Henoc Pérez-Aguilar, Carlos Ruzafa-Silvestre, Elena Orgilés-Calpena, Francisca Arán-Ais

**Affiliations:** Footwear Technological Centre, Campo Alto Campground, 03600 Alicante, Spain

**Keywords:** biomass, cellulose, hemicellulose, lignin, fibers, valorization, lignocellulosic source, circular bioeconomy

## Abstract

This paper briefly discusses the utilization of pruning wastes as a lignocellulosic source of cellulose fibers, which could be of potential use in the development of valuable materials such as sustainable textiles and fillers for footwear components including uppers and soles. *Phoenix canariensis* palm leaves, one of the most common plants found in the local environment of the Alicante region (Spain), was used as a biomass raw material. Determining appropriate processing parameters and their desired range of maximum cellulose extraction states is key to improving yields. Therefore, this study aimed at determining the effect of processing conditions on cellulose extraction by optimizing the hydrothermal process, as a part of overall combined processes involving several steps. Specifically, the time of the steam-explosion stage was varied between 15 and 33 min in order to maximize the cellulose extraction yield. The composition of both the extracted fibers and the resulting by-product solutions generated during the different steps were determined by FTIR and TGA in order to analyze the effectiveness of removing hemicellulose, lignin and extractives as well as the removed substances at each stage for their further valorization. Additionally, the morphology of cellulosic fibers was evaluated by SEM and their crystallinity by XRD. Crystalline cellulose fibers were successfully extracted from pruning biomass wastes, achieving more efficient removal of hemicellulose and lignin when the hydrothermal process was assessed over 25–33 min. This resulted in finer and smoother fibers, but the crystallinity of α-cellulose decreased as the time of steam-explosion increased to 33 min. The characterization of waste solutions generated after the different extraction steps confirmed that the most effective treatments to remove lignin and hemicellulose from the cell wall are alkaline pretreatment and a hydrothermal process.

## 1. Introduction

Currently, the scarcity of natural resources, the high price of energy and the depletion of fossil feedstock have encouraged research into obtaining new sources of energy and alternative renewable raw materials to those derived from petroleum. In this sense, lignocellulosic biomass wastes, such as those generated in pruning or agricultural activities, could be transformed in biorefineries to obtain biocompounds and biofuels of highly interesting industrial applications [1]. Cellulose fibers and nanofibers could be valuable alternative feedstocks to obtain sustainable textiles or biopolymer-reinforced composites, which could contribute significantly to the production of sustainable and low environmental footprint biobased compounds [2,3,4,5,6]. Thus, these wastes constitute a renewable raw material available in large quantities and at low cost, lacking other economically viable applications and whose disposal is necessary from an environmental point of view.

However, due to their complex structure, an adequate pretreatment process is required to develop a suitable biorefinery process.

Lignocellulosic biomass can come from different sources such as agricultural residues in the form of pruning wastes, among others. Depending on the type of lignocellulosic biomass, it usually contains 25–45 wt% cellulose, 20–40 wt% hemicellulose and 10–25 wt% lignin. Cellulose is a homopolysaccharide consisting of linear chains of *D*-glucose subunit bonded by *β*-(1–4) glycoside linkages. Hemicellulose is a heterogeneous branched polysaccharide consisting of galactose, arabinose, mannose, glucose, and xylose. In addition, lignin is an aromatic phenylpropane polymer whose precursors mainly made from coniferyl alcohol, p-coumaryl alcohol and sinapyl alcohol. It has a structural function in the plant cell, binding cellulose and hemicellulose. It also contains extractives of the non-structural components of the biomass such essential oils, fats, waxes, phenolics, and fatty acids, among others [7,8,9,10].

Cellulose is bio-polymerized with a high degree of polymerization, typically 300–1000 (to as high as 10,000–15,000) glucose units in each molecule, giving a molar mass range of 540,000–1,800,000 g/mol. Cellulose thus consists of high molar mass, planar, β-1,4-, cellobiose repeat units, with potential for intramolecular hydrogen bonding, hindrance to rotation within a chain and potential for intermolecular hydrogen bonding—assembling chains into regular arrays, crystals and microfibrils. The structure of cellulose I crystals or native cellulose has adjacent chains in parallel orientation, being the less stable crystalline form of cellulose in terms of thermodynamics. Nevertheless, it is the most abundant native form [11].

Cellulose II occurs after swelling and rearrangement or regeneration of cellulose, although there are instances of natural formation [12].

The extraction of cellulose was assessed by removing the hemicellulose, lignin and extractives from biomass waste. Cellulose can be extracted in many ways. Some chemical, physical, biological and hydrothermal technologies can be used for the pretreatment of plant biomass [1,13,14,15,16,17,18]. Physical pretreatment is mainly carried out to decrease the particle size and increasing the surface area, which will make the subsequent extraction processes more effective and easier [19,20]. According to the literature, lignocellulosic biomass is chemically pretreated using a variety of acids, alkalis, ionic liquids, organic solvents and oxidizing agents. Alkali solutions weaken the sidechains of ester and glycosidic linkages, which change the structure of lignin, cellulose, and hemicellulose. The oxidation uses a variety of oxidizing chemicals to break down hemicellulose and remove lignin. Hydrothermal pretreatment (HTP) is a method in aqueous conditions which combines physical and chemical processes. One of them is steam-explosion, whose mechanism is based on the depolymerization of lignin and explosion of the cellulosic fibrils by treating the biomass to pressurized steam at high temperature, usually ranging from 160 to 280 °C. To extract cellulose fibers, the hydrothermal temperature needs to be below 240 °C to maintain the integrity of the cellulose [1,19,20]. The water contained in the substrate evaporates and swells rapidly, resulting in some degree of hydrolysis.

During steam-explosion, an autohydrolysis mechanism takes places in which the hydrolysis of hemicellulose produces acetic acids and other organic acids, which further help to break down the ester and ether linkages in the matrix. Ester linkages binding carbohydrates and lignin are broken during hydrothermal process causing lignin to melt and recondense [7]. It was demonstrated that hemicelluloses are mainly released in the forms of water-soluble oligomers and monomers during HTP [21]. Concerning lignin, HTP breaks it down into small molecular weight fractions that can repolymerize under severe conditions to form new lignin fragments or more condensed lignin. Reaction temperature, residence time and pressure play key roles in the steam-explosion of biomass. Steam-explosion is widely used to generate organic acids and to reduce sugars and biochemicals, prior to enzymatic pretreatment [1,8,19].

This paper focuses on isolating cellulose fibers from *Phoenix canariensis* palm leaves by purposing an extraction process based on several chemical and physical methods, a steam-explosion process among them. Specifically, the time of this hydrothermal step was optimized in order to maximize cellulose fibers as a biopolymer for potential use to obtain renewable raw materials with high added value for footwear applications.

## 2. Materials and Methods

### 2.1. Materials

Biomass wastes were collected from pruning leaves from ornamental palms (*Phoenix canariensis*), a large palm native to the Canary Islands, Spain. The palms are often used in ornamental landscapes around the world and have grown spontaneously in certain areas of Mediterranean countries, such as the Valencian region, where palm waste [22] from pruning was collected, as shown in Figure 1.

Sodium hydroxide (NaOH), hydrogen peroxide (H_2_O_2_), acetic acid (CH_3_COOH), ethanol absolute and sodium hypochlorite (NaClO) were high grade chemicals, purchased from Merck KGaA (Darmstadt, Germany). All the chemicals and reagents were prepared in distilled Milli-Q water. The chemicals used in this study were employed as received.

### 2.2. The Extraction of Cellulose Fibers from Palm Leaves Waste

The multistep method, based on the combination of several consecutive processes to isolate cellulose fibers, is presented in Figure 2. In this study, in order to optimize the extraction method, the processing conditions of steps 1, 3 and 4 were set and the time of steam-explosion (step 2) was varied to evaluate the influence of this parameter in the effectiveness of the biopolymer extraction.

Initially, the leaves were cut longitudinally to 10 cm in size, as shown in Figure 1b, to improve the area of interaction and to favor contact with the treatment solution.

Step 1: Alkaline pre-treatment. 20 g of cut dried palm leaves wastes was immersed in hot water at 70 °C. The pH was maintained at 12.5 by adding NaOH for 3 h. The waste-to-water ratio was adjusted to 1:20. After treatment, the media were allowed to cool at room temperature and the treated leaves were washed with the de-ionized water, repeatedly, and then filtered. For washing and obtaining neutral fibers, a 10% acetic acid solution was used.

Step 2: Steam-explosion. The alkaline-pre-treated palm leaves were subjected to a hydrothermal process, using a high temperature/high pressure lab reactor (HPHT) (Novoclave, Büchi AG—Pilot Plant & Reactor Systems; Uster, Switzerland). The ratio of water:vegetable residue was 10:1, the reactor was closed and heated by means of its thermal jacket at a temperature of 200 °C inside the reactor (Figure 2), without allowing the release of pressure. The reactor was left at 200 °C at different processing times to evaluate the influence of time on the effectiveness of the depolymerization of lignin and degradation of hemicellulose. Once this treatment had been finalized, the reactor was depressurized and allowed to cool to room temperature overnight. The time of steam pre-treatment was optimized by varying from 15–33 min, according to previous studies carried out by the authors.

Step 3: Oxidation treatment: After the hydrothermal process, the remaining solid was filtered and treated with a solution of hydrogen peroxide at 2 wt% (ratio 20:1). The temperature was fixed at 50 °C and pH at 11.5 for 5 h. To facilitate the purification of the fibers, they were washed with distilled water, water: ethanol (1:1), and finally, with absolute ethanol.

Step 4: Bleaching process. The process consists of treating the extracted cellulose with an aqueous solution of NaClO 10 wt% to remove lignin and amorphous cellulose. The ratio of solution:fibers was 15:1. The solution was stirred for 1 h at a fixed temperature (70 °C). The resulting product was filtered and neutralized by washing it with water repeatedly. Finally, the obtained fibers were dried at 80 °C in an oven until a constant weight was reached.

### 2.3. Experimental Techniques

The characterization of the cellulose fibers and the waste solutions generated during the different steps has been carried out by:

#### 2.3.1. Fourier Transform Infrared Spectroscopy (FTIR)

A Varian 660-IR infrared spectrophotometer (Varian Australia PTY LTD; Mulgrave, Australia) was used to determine the composition of the cellulosic fibers. The Attenuated Total Reflection (ATR) technique was used by carrying out 16 scans with a resolution of 4 cm^−1^ [23].

#### 2.3.2. Thermogravimetric Analysis (TGA)

A TGA 2 STARe System thermal balance with STARe software (Mettler-Toledo; Columbus, Ohio, USA) was used to test the thermal stability of cellulose and solutions. In an alumina crucible, 5 to 7 mg of the sample was added. The sample was heated from 30 to 600 °C at 10 °C/min, using a nitrogen flow rate of 30 mL/min [22]. Each sample was analyzed in triplicate, the results shown are an average of the three tests performed. The error tests were less than ±0.05%.

#### 2.3.3. X-ray Diffraction (XRD)

To determine the quantitative analysis of crystalline fibers, a non-destructive technique was employed. A wide angle XRD on a Bruker (Billerica, Massachusetts, USA) D8-Advance Göebel mirror (non-flat samples) with a high temperature chamber (up to 900 °C) and a Siemens Bruker Kristalloflex K 760–80F X-ray generator (Power: 3000 W, Voltage: 20–60 KV and Current: 5–80 mA) equipped with an XR tube with copper anode was used to analyze the crystallinity [24].The crystallinity index was calculated using the peak height method and Equation (1) [25]. Moreover, the percent crystallinity was calculated using Equation (2).
(1)$$CI=\frac{\left(I_{002}-I_{am}\right)}{I_{002}}\times 100$$
(2)$$\%\;Cr=\frac{I_{002}}{\left(I_{002}+I_{am}\right)}\times 100$$

In both equations, *I*_002_ is the peak intensity at *2θ* = 22° and *I_am_* is the peak intensity at *2θ* = 16°.

#### 2.3.4. Elemental Analysis (EA)

To determine the quantitative elemental analysis of carbon, hydrogen, nitrogen and sulphur of cellulose isolated fibers, a Thermo Finnigan LLC (San Jose, CA, USA) Flash 1112 Series elemental microanalyser was used. The elemental CHNS microanalyser was equipped with a LECO Corporation (St. Joseph, MI, USA) Micro TruSpec detection system. It uses a combination of continuous flow of helium carrier gas, infrared detector and thermal conductivity detector giving simultaneous detection of CHNS.

#### 2.3.5. Scanning Electron Microscopy (SEM)

A Phenom ProX scanning electron microscope (Phenom World, Eindhoven, The Netherlands) was used for the morphological study. The microscope uses an electron beam with a potential of 5 to 15 keV and operates in a high vacuum to increase image resolution [26].

#### 2.3.6. Water Contact Angles (WCA) Measurements

Static water contact angles were obtained with an Attention Theta Flow optical tensiometer (Biolin Scientific Oy, Espoo, Finland). Measurements were performed using three drops of double distilled water as measured according to EN 828:2013 on a rectangular sample of isolated cellulose fibers to evaluate the hydrophilicity and hydrophobicity of the materials studied [27,28].

#### 2.3.7. Determination of the Static Absorption of Water

This test method is based on standard ISO 2417:2016 [29]. The test procedure requires a piece of known mass of the palm sample to be immersed in water for a given period of time, in this case 5 min, and the mass of water absorbed by the sample is then determined.

## 3. Results and Discussion

### 3.1. Characterization of Cellulose Fibers

Chemical properties of the extracted cellulose fibers from palm leaves waste as a function of the steam-explosion treatment time were analyzed by FTIR spectroscopy. The FTIR spectra of extracted cellulose fibers as well as the palm pruning waste are shown in Figure 3. The most characteristic peaks of common functional groups in biomass clearly demonstrated the abundance of cellulose, hemicellulose and lignin in the samples (Table 1).

The spectrum corresponding to the palm pruning waste is quite similar to the extracted fibers at lower hydrothermal times (15 and 25 min). They show a broad absorption band at around 3390 cm^−1^, which corresponds to the O–H *st* in cellulose. The peak at 1727 cm^−1^ is indicative of C=O and C–O *st*, respectively, for the hemicelluloses and lignin as well as the extractive compounds. The peak at 1643 cm^−1^ was associated with water absorption into fibers. The absorption band at 1600–1500 cm^−1^ corresponds to the C=C aromatic skeletal vibration of lignin. The peak at 1434 cm^−1^ corresponds to the asymmetric CH_2_ bending attributed to the crystalline cellulose. The bands at 1161, 1108, 1055 and 1034 cm^−1^ are related to C-O skeletal vibration of cellulose and hemicellulose and the band located at 900 cm^−1^ to glucose ring stretching (C1–H) deformations of cellulose and hemicelluloses.

According to the FTIR spectra, as hydrothermal time increases and the removal of the lignin and hemicelluloses takes place, this is characterized by a decrease or disappearance of the band associated with C-H *st* (indicative of a reduction of extractives such as waxes or oils), the band at 1727 cm^−1^ (characteristic of C=O *st* of carbonyl related to hemicelluloses and lignin) and the band at 1519 cm^−1^ (corresponding to the C=C aromatic skeletal vibration of lignin). Furthermore, the bands at 1161, 1108, 1055 and 1034 cm^−1^ due to C-O skeletal vibration of cellulose, as well as the intensity of the band related with cellulose located at 900 cm^−1^, increased as a result of the extracted fibers, which was more noticeable when steam-explosion was fixed at 33 min. According to the results, there was a significant decrease regarding the lignin content, hemicellulose and extractive compounds mainly observed at 33 min, which confirms that the cellulose content increased significantly after chemically treating the fibers [30].

Furthermore, TGA has been used to determine the chemical composition and content in terms of α-cellulose, lignin and hemicellulose with enhanced accuracy compared to commonly used wet chemical methods from different plants [34,35] and its high-throughput nature (at-line). Therefore, this method is ideal for assessing polymers such as lignin.

Figure 4a shows the thermogravimetric weight loss curves of the biomass waste as well as the extracted fibers obtained by TGA. Cellulose, hemicellulose and lignin show different thermal decomposition profiles, probably because of differences in their inherent structures and chemical natures. Hemicellulose is composed of various saccharides (xylose, mannose, glucose, galactose, etc.). Its random, amorphous and branched structure facilitates its decomposition into volatiles (CO, CO_2_, and some hydrocarbon, etc.) at low temperatures (approximately between 200–300 °C). Different to hemicellulose, cellulose is a polymer composed of glucose units attached together by β-1,4 glycosidic bonds that confer strength and thermal stability. According to bibliography, cellulose decomposes in the range of 300–370 °C. Among the three components, lignin has the higher thermal stability. This is due to the fact that lignin contains a high content of aromatic rings with different branches, so the decomposition profile occurs in a wider temperature range (200–500 °C) than for cellulose and the hemicellulose components of biomass. Additionally, the weight loss observed below 110 °C corresponds to the moisture bonded to the hydroxyl and carboxyl groups of the fibers [14,36].

Differential Thermal Analysis (DTA) curves, obtained from the first derivative of TGA curves corresponding to the derivative curve based on time, are shown in Figure 4b. The curve corresponding to biomass waste showed a broad and multi loss-weigh profiles due to its lignocellulosic composition. The decomposition process depends on the content of these main components, so the degradation profile results from the degradation of cellulose, complemented by a shoulder at low temperature, associated with hemicellulose degradation, and a tail at high temperature corresponding to lignin decomposition [14,36,37]. The removal of the hemicellulose from the palm waste promoted by the chemical treatment is confirmed by the disappearance of the shoulder/peak observed at 281 °C in the DTG curve. Apart from that, lignin removal is also confirmed by the reduction of the amount of residue in the range of 370–500 °C. All extracted fibers show a sharp and clear peak, the temperature at maximum loss-rate taking place at 381 °C for 15–25 min of steam-explosion, whereas it decreased to a lower temperature (330 °C) when the explosion time was fixed at 33 min. This confirms the removal of both lignin and hemicellulose from the raw material, although the sample shows a small broad peak over 400 °C at 33 min, which corresponds to a small amount of lignin not completely released from the cell wall.

This difference in decomposition temperature at 33 min showed that the temperature interval is narrower than the corresponding intervals at 15 and 25 min. This means that the cellulose fiber at 33 min has less impurities corresponding to hemicellulose, lignin or their decomposition products that remain attached and make the cellulose peak wider as it occurs at 15 and 25 min, causing an increase in temperature at maximum loss-rate. However, this does not mean that the fibers isolated at 15 and 25 min are more thermally stable than those obtained at 33 min. Otherwise, it implies that cellulose fibers at 15 and 25 min have more impurities corresponding to the degradation of lignin and hemicellulose, which shifts the decomposition temperature of the cellulose to a higher temperature value.

These results are in line with the previously shown FTIR spectra, which show an increase in the intensity of the band corresponding to the OH stretching of cellulose (3600–3100 cm^−1^) with increasing steam-explosion time from 15 to 33 min, as well as a loss of intensity of the C-H stresses of the aromatic rings corresponding to lignin (2990 and 2850 cm^−1^). The disappearance of the carbonyl corresponding to the hemicellulose (1733 cm^−1^) at 33 min is also observed. Finally, the peaks corresponding to the carbon structure of the cellulose (1160–1000 cm^−1^) are more intense when the process is carried out for 33 min, which proves that this sample is mostly composed of cellulose fibers and hemicellulose and lignin have been almost completely removed.

Table 2 compiles the results obtained from the differential thermogravimetric analysis (TGA) in which the initial composition of pruning waste corresponding to moisture (peak 1, 94.7 °C), hemicellulose content (peak 2, 281 °C), cellulose (peak 3, 335 °C) and lignin content (peak 4, 410 °C) are included. The results show that the peak corresponding to cellulose in pruning waste and the process at 33 min produce a maximum of decomposition at the same temperature (335 and 338 °C, respectively). However, at 15 and 25 min, the cellulose peak appears at 361 °C, which proves that it is not pure cellulose, but it has rather been contaminated with lignin decomposition products.

The cellulose content of the overall process was included in Table 3. Its calculation was based on the number of extracted fibers at different processing times during the hydrothermal process (according to TGA data).

The XRD patterns of isolated cellulosic fibers are presented in Figure 5. The X-ray diffractograms show four main peaks around 2*θ* = 16°, 22° and 35° corresponding to the (110), (200), and (004) crystal planes of the cellulose I structure, respectively [37,38,39,40,41,42,43]. The alpha-cellulose represents undegraded higher-molecular-weight cellulose, which is the highest quantity and also further used to make high-quality nanocellulose materials [14]. This indicates that the chemical treatments successfully remove lignin, hemicellulose and other non-cellulosic compounds in which they generally exist as an amorphous phase, especially lignin, and guarantees success in extracting the crystalline cellulose phase. Applying 15 min, 25 min and 33 min of hydrothermal treatment had a crystallinity index (C.I.) of 45.2%, 53.9% and 54.0%, respectively, and a crystallinity percentage (% Cr) of 64.6%, 68.4% and 68.5%, respectively. The increase in C.I. and % Cr after increasing the time of hydrothermal treatment is due to the effective removal of the amorphous constituents and improved order of cellulose crystals at the fiber axis. The C.I. of fibers in this work is comparable to other natural fibers such as bagasse, pineapple, coir or jute [44]. The high pressure and given temperature causes swelling of lignocellulose and the rapid release of pressure disrupts the fibrous structure of biomass, reducing the crystallinity of cellulose, which is used in biorefineries to improve the accessibility of enzymes [20]. The increase of crystallinity is in line with a decrease of decomposition temperature of cellulosic fibers extracted at 15 and 25 min, which is proved by TGA results [7,14,42].

In addition, Table 4 shows the elemental analysis of the samples extracted at different extraction times by steam-explosion.

The results show how the nitrogen content (0.13%) is reduced by half as a result of the steam-explosion treatment when the time of the steam-explosion process is increased from 25 min to 33 min. The carbon content (47.37%) and the hydrogen content (6.88%) are also significantly reduced, showing the efficiency of the process to decompose the cell wall of the palm tree leaf and isolate cellulose fibers.

SEM images showing the morphology of the extracted cellulose fibers are included in Figure 6. The palm leaves image shows the structure of the leaf prior to treatment, revealing the cell wall formed by hemicellulose and lignin that protects cellulose. Nevertheless, the image corresponding to leaves before 15 min of steam-explosion shows the leaves after being treated only with steam-explosion cracking the cell wall and with its structure degraded.

The SEM images corresponding to 15 min, 25 min and 33 min show the efficiency of the multistep process to eliminate lignin and hemicellulose from biomass. All photographs prove the presence of cellulosic fibers, which indicates their successful extraction. However, this effect is more noticeable after 25–33 min of hydrothermal process, showing both samples (25 and 33 min) a more complete defibrillation of the microfibrils into individual fibrils. In fact, the cellulose fibers in the sample at 15 min were thicker, whereas the sample at 25 min and 33 min resulted in a substantial reduction of the diameter of the fibrils. Furthermore, as time of exposure to the steam-explosion increased, the surface of the cellulose fibers became more uniform and smoother [45]. In addition, the photographs show how lignin and hemicellulose at 15 min have not been completely removed throughout the sample. However, at 25 min, more uniformity in the fibers was observed, but no lignin polymers were present, although some traces of polymer residue were observed. Finally, in 33 min of steam-explosion, the polymer removal was completed and the isolated cellulose fibers were totally clean and more uniform in both size and diameter than those cellulose fibers isolated at 25 min of the process.

This fact is also in accordance with the color observed in the cellulose samples. As can be seen in Figure 7, the fibers bonded by hemicellulose and lignin resulted in a brown color, but after the multistep extraction process and, specifically, increasing the time of steam-explosion up to 25 and 33 min, the resulting fibers were yellow and white, respectively. The change in color confirms that time processing of steam-explosion is a crucial parameter to wash out hemicellulose and lignin.

Furthermore, the hydrophobicity of the extracted fibers was analyzed by contact angle measurements with double distilled water as the standard liquid. The results obtained from the static contact angle showed that water droplets are instantaneously absorbed by the isolated cellulose fibers obtained at different steam-explosion process times. Therefore, the contact angle of the isolated cellulose fibers is zero, as shown in Figure 8. The isolated cellulose fiber has an outstanding hydrophilic behavior due to a large number of hydroxyl groups in the chain. The contact angle is a quantitative measure of the wettability of a material’s surface. In the case of water as testing liquid, substrates with a contact angle of < 90° are considered to be hydrophilic [46]. In terms of energetics, this implies that the forces associated with the interaction between the water and the surface are greater than the cohesive forces associated with bulk liquid water.

Once the hydrophilicity of the cellulose-isolated fibers had been evaluated, their water uptake capacity was assessed. The results are shown in Table 5.

The test results obtained according to water absorption show that cellulose fibers isolated at a longer steam-explosion time (33 min) have a higher water-absorption capacity. Nevertheless, the water absorption by the cellulose fibers is significantly lower at lower steam-explosion times due to the residual presence of hemicellulose or lignin around the fibers (15 and 25 min).

### 3.2. Characterization of the Generated Waste Solutions at Each Extraction Stage

In addition, the different byproducts obtained at each fiber extraction stage were analyzed. First of all, the dry matter of the waste solutions, which ranged from 24–30 wt%, was determined and is included in Table 6.

After that, the composition of each waste solution was analyzed by FTIR and TGA after drying. Figure 9 displays FTIR spectra obtained from the solutions after the different processes. The spectra are complex because they are not only constituted by hemicellulose, lignin and extractives, but also by the degraded compounds as a consequence of the depolymerization of lignin and hemicellulose. The broad band at 3326 cm^−1^ is associated with the stretching of OH groups on hemicellulose, lignin aliphatic and phenolic -OH groups as well as depolymerized sugars. The bands at 2935 cm^−1^ and 2840 cm^−1^ are related to the C-H methyl and methylene groups stretch vibrations. The band at 1454 cm^−1^ is also associated with the bending deformation of the C-H group. The band at 1700 cm^−1^ is linked to the C=O stretch. The absorption peaks at 1733 cm^−1^ correspond to the C=O *st* of unconjugated ketone, carboxyl, and ester groups, which are characteristic of hemicellulose, organic acids and ester contained in extractives and other resulting degraded compounds. The band at 1600–1587 cm^−1^ is typical of C=C aromatic vibration characteristic of lignin. The highest intense band at 1100 cm^−1^ corresponds to C–O deformation of the secondary alcohols and aliphatic ethers [31].

According to the spectra, the bands typical of lignin and hemicellulose are intensified in the waste solution obtained after alkaline pre-treatment and hydrothermal process. It indicates the effective extraction of hemicellulose, lignin and extractives at these different processes and how each one contributes to successfully remove these substances. Although all steps eliminate lignin, steam-explosion and alkaline pre-treatment seem to be the most effective processes to remove lignin from biomass.

Degradation profiles of species contained in each waste solution were determined by TGA (Figure 10). The peaks < 100 °C correspond to moisture. All curves show peaks lower than the typical degradation temperature of hemicellulose at 285 °C, confirming that hemicellulose, composed mainly of xylan, was solubilized during hydrothermal process, and recovered as soluble saccharides including xylo-oligosaccharides, xylose, and arabinose in the liquid [17,47]. During steam-explosion, the ester bonds that hold lignin and carbohydrates together were broken, causing lignin to melt and then condense. The degradation peak over 400 °C was attributed to lignin and recondensed structures [1,48,49]. Therefore, the solution wastes after the alkaline pre-treatment and steam-explosion show a broad peak over 500 °C, which confirms mainly the removal of lignin in both of these stages. The subsequent oxidation treatment allows the removal of the lignin structure, which was probably degraded in the previous stages as a lower degradation temperature (<500 °C) is evident. Cellulose is largely preserved in its original form, and only slight solubilization occurs with steam-explosion since the solution after steam-explosion shows a peak at 330 °C (characteristic of cellulose).

## 4. Conclusions

Cellulose fibers were successfully extracted from pruning palm leaves waste using a four-step method consisting of an alkaline pre-treatment, a hydrothermal process (steam-explosion), oxidation treatment and bleaching. In this study, the influence of steam-explosion times on the effectiveness of the biopolymer extraction was evaluated. The extraction process defibrillates the cellulose bundles, resulting in individualized microfibers, more noticeable after 25–33 min of the hydrothermal process. The morphological surface shows that extracted cellulosic fibers are finer and smoother after 25–33 min of steam-explosion as a consequence of removing hemicellulose and lignin from the cell wall; this was confirmed by the analysis of the chemical composition of the resulting fibers. XRD analysis indicated the crystalline structure of the *α*-cellulose; an increase of crystallinity was observed as the time of steam-explosion increased to 25–33 min. Finally, the characterization of the waste solutions generated after the different extraction steps confirmed that alkaline pre-treatment and the hydrothermal process seem to be the more effective treatments to remove lignin and hemicellulose, which is solubilized into lower molecular weight compounds during steam-explosion.

These results offered a crucial outlook on the production of cellulose fibers from plentiful trimmings of palm leaf waste, which can be profitably used as a valuable feedstock to obtain an alternative to cellulose from cotton for sustainable textiles, nanocellulose or to reinforce a composite biopolymer matrix. This could contribute significantly to reducing the environmental footprint of several industries, such as the footwear sector. Therefore, the development of innovative materials for footwear applications based on such fibers will be the objective of further research carried out by the authors.

## Figures and Tables

**Figure 1 polymers-14-05206-f001:**
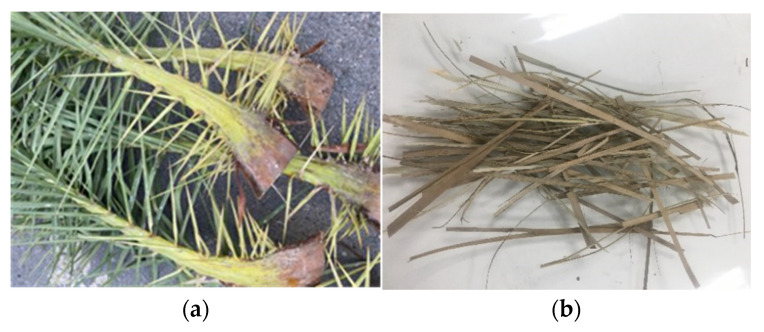
Palm leaves wastes (**a**) fresh palm leaves and (**b**) cut dried palm leaves.

**Figure 2 polymers-14-05206-f002:**
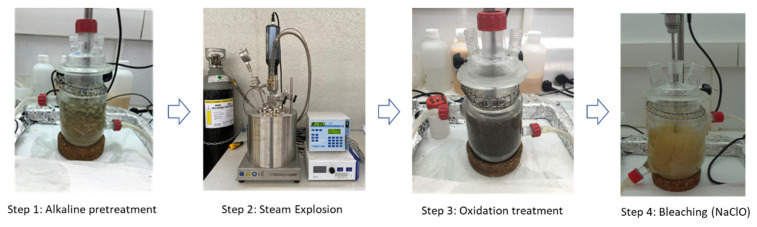
Extraction process to obtain cellulose fibers from palm leaves waste.

**Figure 3 polymers-14-05206-f003:**
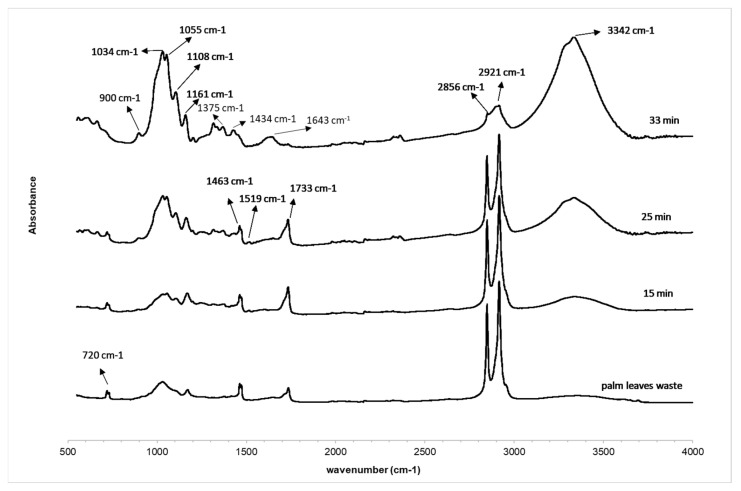
FTIR spectra of palm leaves waste and the cellulose fibers extracted at different times of steam-explosion treatment.

**Figure 4 polymers-14-05206-f004:**
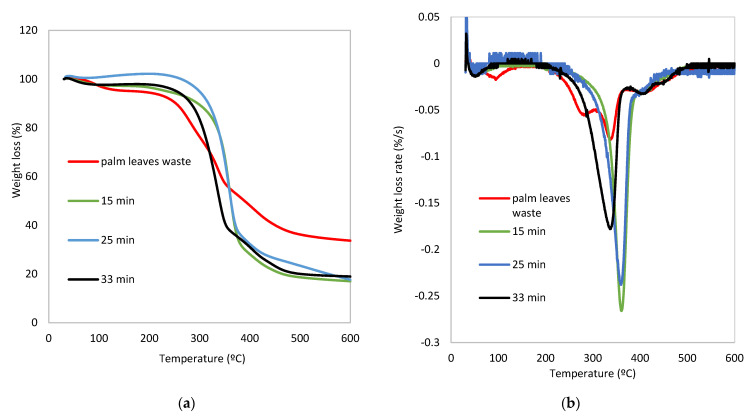
(**a**) TGA and (**b**) DTA curves corresponding to pruning waste and the resulting fibers.

**Figure 5 polymers-14-05206-f005:**
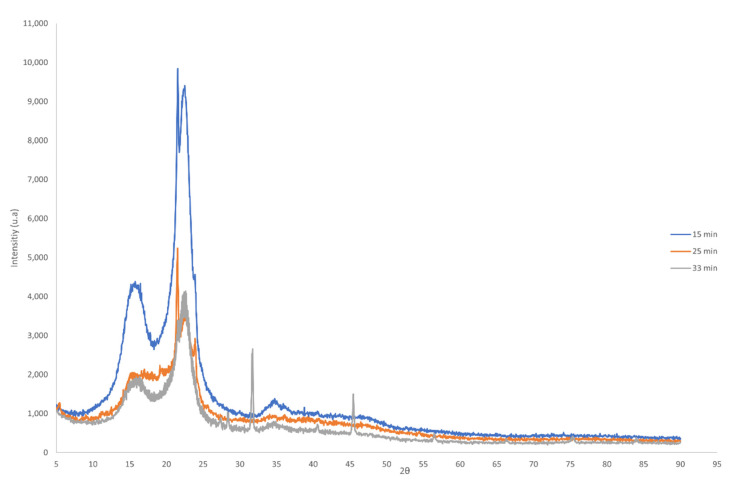
XRD profile of extracted fibers at different time of hydrothermal step.

**Figure 6 polymers-14-05206-f006:**
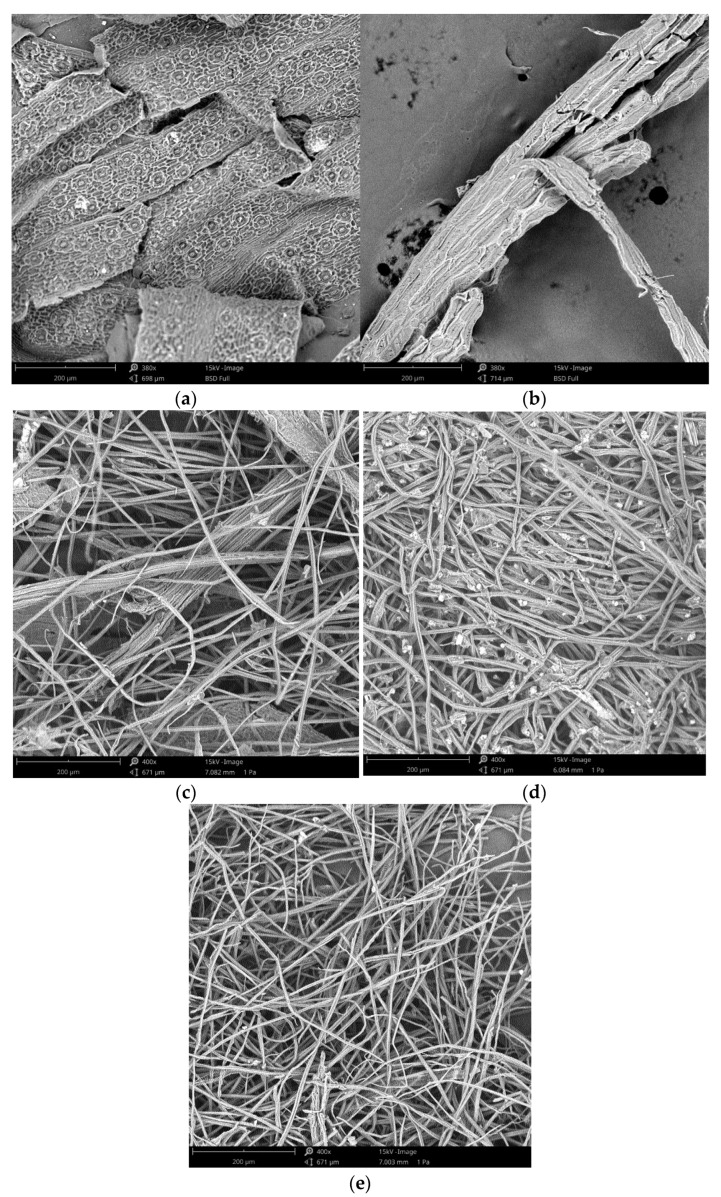
SEM images of fibers extracted at different steam-explosion times. (**a**) Untreated palm leave, (**b**) Leaves before 15 min of steam-explosion, (**c**)15 min, (**d**) 25 min, (**e**) 33 min.

**Figure 7 polymers-14-05206-f007:**
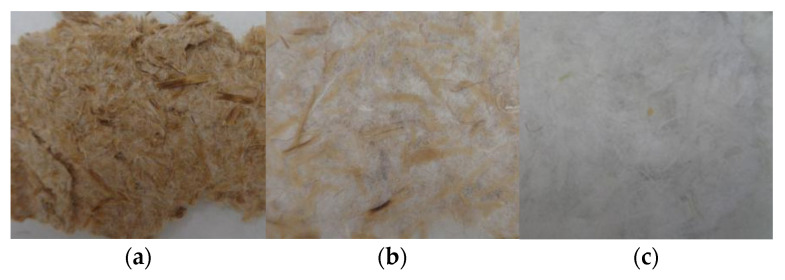
Cellulose fibers extracted with different conditions resulting from the steam-explosion process (Step2). (**a**) 15 min, (**b**) 25 min, (**c**) 33 min.

**Figure 8 polymers-14-05206-f008:**
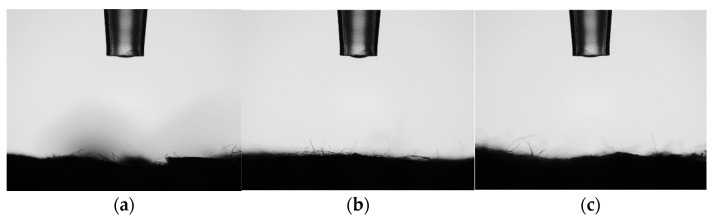
Static water contact angles tests for cellulose extracted at different steam-explosion times (15 min, 25 min and 33 min). (**a**) 15 min, (**b**) 25 min, (**c**) 33 min.

**Figure 9 polymers-14-05206-f009:**
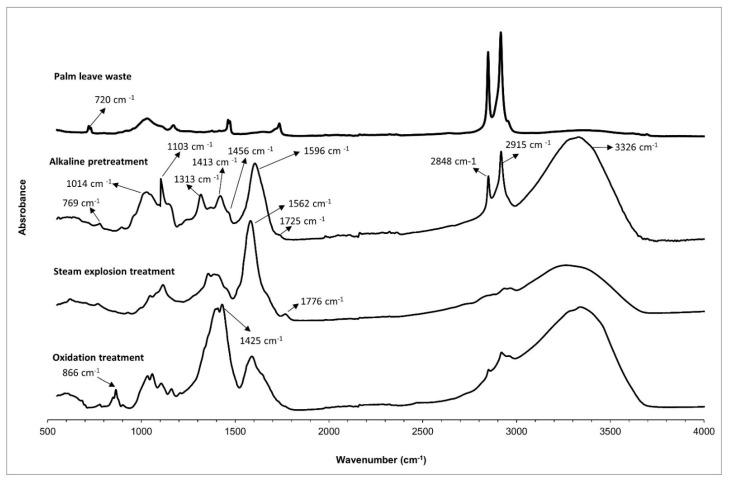
FTIR spectra of the dried waste generated at each step.

**Figure 10 polymers-14-05206-f010:**
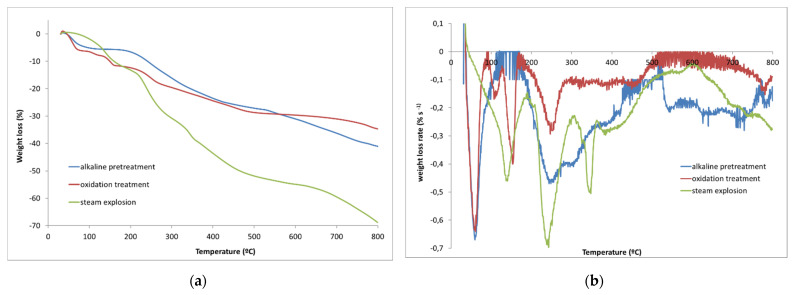
Thermograms of dried waste solutions generated after alkaline pre-treatment, hydrogen peroxide treatment and steam-explosion (**a**) TGA; (**b**) DTA.

**Table 1 polymers-14-05206-t001:** Infrared bands assignments for cellulose, hemicellulose and lignin [31,32,33].

Wavenumber (cm^−1^)	Assignment
3600–3100	Stretching of OH groups on cellulose, hemicellulose and lignin
2990–2850	C-H stretching vibration
1733	C=O stretching of carbonyl related to hemicelluloses, lignin and extractives
1643	Absorbed water
1600–1500	Aromatic skeletal vibration corresponds to the presence of pure hemicellulose and lignin
1463	C-H_2_ *δ* in lignin, carbohydrates and waxes
1434	A symmetric CH2 bending vibration attributed to crystalline cellulose
1375	C-H_3_ *δ* *sy*
1161	C-O asymmetric bending cellulose
1108	C-OH skeletal vibration
1055	C-O-C skeletal vibration
1034	C-O stretching vibration
900	*β*-glycosidic C-H deformation cellulose
720	(CH_2_)_n_ out of plane deformation *n* > 4 (oils and waxes)

**Table 2 polymers-14-05206-t002:** Data obtained from DTA curves.

	Peak 1		Peak 2		Peak 3		Peak 4		
Sample	T (°C)	Weight-Loss (%)	T (°C)	Weight-Loss (%)	T (°C)	Weight-Loss (%)	T (°C)	Weight-Loss (%)	Residue (wt%)
Pruning waste	94.7	4.8	281	20.8	335	20.6	410 °C	20.1	33.7
15 min					361	68.8			17.0
25 min					361	67.7			17.7
33 min					338	56.6	380–500	14.6	19.0

**Table 3 polymers-14-05206-t003:** Cellulose content calculated from DTA data as a function of the steam-explosion stage time.

Time	Cellulose Content (wt%) ^1^
Waste	4.8
15 min	68.8
25 min	67.7
33 min	56.6

^1^ Data from DTA. Obtained from the weight loss peak at 330–361 °C.

**Table 4 polymers-14-05206-t004:** Elemental composition of fibers extracted at different steam-explosion times.

Time	Nitrogen %	Carbon %	Hydrogen %	Sulfur %
15 min	0.29	50.67	7.18	0.00
25 min	0.29	57.31	8.35	0.00
33 min	0.13	47.37	6.88	0.00

**Table 5 polymers-14-05206-t005:** Static water absorption tests for cellulose fibers isolated at different steam-explosion times (15 min, 25 min and 33 min).

Steam-Explosion Time	Water Absorbed (wt%)
15 min	701.5
25 min	897.5
33 min	1561.0

**Table 6 polymers-14-05206-t006:** Dry matter content of generated solutions after each step.

Step	Dry Matter (wt%)
Alkaline pretreatment	24.11
Steam-explosion	20.36
Oxidation treatment with H_2_O_2_	30.18

## Data Availability

The data presented in this study are available on request from the corresponding author.

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
