# Peer review of "Effect of Processing Time of Steam-Explosion for the Extraction of Cellulose Fibers from Phoenix canariensis Palm Leaves as Potential Renewable Feedstock for Materials"

_polymers, 2022, doi:10.3390/polym14235206_

Round 1

Reviewer 1 Report

The manuscript describes the extraction technique of cellulose fibers from palm leaves for footwear materials manufacturing. The obtained data are valuable in this scientific area. This manuscript can be accepted. But there is an issue that need revision.

1) Why didn't you analyze the hydrophilicity of the isolated fibers, e.g., by measuring water uptake and the water contact angle?

Author Response

Please find attached the relevant document.

Reviewer 2 Report

The main objective of this work was to obtain cellulose fibers from Phoenix canariensis palm leaves wastes. The focus of the study was on the time adopted during one of the cellulose extraction steps, which is the steam explosion (3 different times were used). The characterization of treated fibers was performed by SEM, FTIR, TGA and XRD. The residue generated after each extraction step was also evaluated. This study is important, since it shows the influence of treatment conditions to obtain cellulose from biomass, in a more sustainable way compared to treatments with acids. However, article needs a major revision, according to the following comments:

Introduction:

The introduction topic is well written. The structure of cellulose should be detailed, including some definitions that are cited in the discussion of results (such as cellulose I, for example).

Materials and Methods:

-       Figure 1 was not cited in the text; also, specify the difference between the two images, or keep only one of them.

-       Were the pruning leaves used as received, or did they pass through some process (such as milling, for example) before Step 1? All steps must be detailed.

-       Specify how many TGA test were performed for each sample.

Results and Discussion:

-       SEM image of the fibers before treatment should be included (before Steam Explosion, for example) to enhance discussion of the effectiveness of the process. Also, specify how it is possible to conclude from SEM images that lignin and hemicellulose were eliminated. 

-       The mean diameter of fibers treated for 33 min appears larger compared to fibers treated for 25 min. Include a comment about this. It is important to improve the scale bar definition in figure 3.

-       Figure 4: Include some dimension reference.

-       Correlate FTIR results with similar results observed in the literature.

-       TGA: Table 2 was not cited in the text. Explain in more detail the reason for the decrease in thermal stability for 33min treatment. It was also not clear the correlation of the result at this temperature with the FTIR. On page 8 (before figure 6): reference to “sample C” must be modified, specifying the treatment time. Was the first derivative of the TGA curve based on temperature or time? How accurate is the cellulose content result obtained from the TGA test?

-       Some chemical analysis must be carried out on treated fibers to know the exact content of cellulose, hemicellulose and lignin. This allows for a more in-depth discussion of results.

-       Figure 7: Include the results for the pruning waste. If possible, shift the curves (as in figure 5).

-       Specify how cellulose crystallinity is evaluated from the X-ray diffraction results; eventually calculate this parameter.

Final comment:

-       The title of the article refers to the use of cellulose fibers as a material for footwear. However, no explanation of this specific usage is made in the article. Therefore, I recommend an adjustment on title, or the addition of information on the use of this fiber in the footwear industry.

Author Response

(The authors gave the same response as above.)

Round 2

Reviewer 2 Report

The article quality improved after review. Some additional analyzes were also carried out to complement results discussion. All suggested modifications were implemented and commented in the text. As a suggestion for future work, I recommend using a more accurate method to determine the cellulose percentage.

I recommend, therefore, the publication of the work.